# Improving Value Estimation Critically Enhances Vanilla Policy Gradient

Tao Wang [1]   Ruipeng Zhang [1]   Sicun Gao [1]

## Abstract

Modern policy gradient algorithms, such as TRPO and PPO, outperform vanilla policy gradient in many RL tasks. Questioning the common belief that enforcing approximate trust regions leads to steady policy improvement in practice, we show that the more critical factor is the enhanced value estimation accuracy from more value update steps in each iteration. To demonstrate, we show that by simply increasing the number of value update steps per iteration, vanilla policy gradient itself can achieve performance comparable to or better than PPO in all the standard continuous control benchmark environments. Importantly, this simple change to vanilla policy gradient is significantly more robust to hyperparameter choices, opening up the possibility that RL algorithms may still become more effective and easier to use.

## 1. Introduction

Deep policy gradient methods have demonstrated their strength in various domains, including robot learning (Song et al., 2023), game playing (Vinyals et al., 2019), and large language model training (Ouyang et al., 2022). Policy gradient methods in its modern versions, such as Trust Region Policy Optimization (TRPO) (Schulman et al., 2015a), Proximal Policy Optimization (PPO) (Schulman et al., 2017), have been the popular choice for delivering state-of-the-art performance across the broad range of RL tasks. In comparison, the original policy gradient algorithm, Vanilla Policy Gradient (VPG) (Sutton et al., 1999), typically performs significantly worse than the modern algorithms. The common explanation for why TRPO and PPO outperform VPG is their ability to prevent excessive large policy updates. This is achieved through a surrogate objective that incorporates importance sampling, constrained by a limit on policy step size each iteration. While monotonic im-

[1]University of California, San Diego, La Jolla, USA. Correspondence to: Tao Wang <taw003@ucsd.edu>.

*Proceedings of the 42nd International Conference on Machine Learning*, Vancouver, Canada. PMLR 267, 2025. Copyright 2025 by the author(s).

provement is theoretically guaranteed for sufficiently small policy updates (Kakade & Langford, 2002; Schulman et al., 2015a), empirical results suggest otherwise: smaller learning rates for the policy network do not always yield better performance (Andrychowicz et al., 2021). According to the trust region theory (Schulman et al., 2015a), there is a gap between theory and practice because the theory guarantees improvement of the original policy objective only when the surrogate objective improves by a margin greater than the distance between the old and new policies. However, there is no guarantee that this condition is consistently satisfied in practical policy training.

Moreover, it has been reported that deep policy gradient methods often suffer from implementation issues such as brittleness, poor reproducibility, and sensitivity to hyperparameter choices (Henderson et al., 2018; Engstrom et al., 2020; Andrychowicz et al., 2021). Extensive empirical analysis has shown that the behavior of deep policy gradient methods does not align with the predictions of their motivating framework (Ilyas et al., 2020). Recent studies further highlight that the optimization landscape of many RL tasks is highly non-smooth and even fractal, raising questions about the well-posedness of all policy gradient methods (Wang et al., 2023). Consequently, there is still no clear answer to the question: *while almost all theoretical assumptions are violated in practice, why does PPO perform better than vanilla policy gradient empirically?*

In this paper, we present a different perspective on deep policy gradient methods within the actor-critic framework: accurate value estimation is more critical than enforcing trust regions. We first demonstrate that, in practice, trust region methods do not behave as their theoretical analysis suggests. Then, we analyze how TRPO and PPO implicitly enhance value estimation through their implementation designs. Our theoretical analysis reveals that the true value often grows significantly faster than the critic's predictions, leading to poor value estimation during policy training. Finally, empirical results show that simply increasing the number of value updates enables the basic VPG algorithm to match PPO's performance across multiple continuous control benchmarks in Gymnasium. This highlights the pivotal role of value estimation in improving policy gradient methods.

This paper is organized as follows. In Section 4, we analyze

the relationship between trust regions and policy improvement, demonstrating that there is no clear correlation. This suggests that enhancing a trust region may not be the fundamental reason for the success of trust region methods. In Section 5, we explain that both TRPO and PPO perform more value steps per iteration compared to VPG, fundamentally contributing to closer value approximation and improved performance. Additionally, we provide a theoretical framework suggesting that value networks typically require more gradient steps to optimize than policy networks. In Section 6, extensive experiments are conducted and presented to corroborate our theoretical analysis. Specifically, we show that by increasing the number of value steps alone, VPG achieves performance similar to or better than PPO across several Gymnasium benchmarks.

## 2. Related Work

**Performance gap between VPG and PPO.** Much work has been done to understand the performance gap between VPG and PPO. It has been shown that the VPG loss performs significantly worse than the PPO loss, even with optimally conditioned hyperparameters (Andrychowicz et al., 2021; Raffin et al., 2021). On the theoretical side, the global optimality of TRPO/PPO is proven for overparameterized neural networks under the assumption of a finite action space (Liu et al., 2019). It is also shown that ratio clipping may not be necessary in PPO (Sun et al., 2023). Another theory suggests that policy gradients work by smoothing the value landscape (Wang et al., 2024), indicating that the choice of policy objective may not be a critical factor affecting performance, as all of them provide the same smoothing effect through Gaussian kernels. In this work, we take a step forward to demonstrate that the fundamental gap between VPG and PPO lies in value estimation, and by optimizing it, VPG can achieve performance comparable to PPO.

**Value estimation in off-policy methods.** Although it has been demonstrated that an optimal baseline in policy gradients can effectively reduce the variance of the policy gradient estimator (Peters & Schaal, 2008), it remains unclear to what extent value estimation affects the performance of on-policy algorithms. In contrast, the importance of value estimation has been extensively studied in off-policy algorithms. In particular, it has been found that the max operator may lead to overestimated $Q$ values, and double $Q$ learning methods were introduced to mitigate this error in estimation (van Hasselt, 2010; van Hasselt et al., 2016). This method also serves as one of the keystones in advanced off-policy algorithms such as TD3 (Fujimoto et al., 2018) and SAC (Haarnoja et al., 2018), compared to the DDPG algorithm (Lillicrap et al., 2015). Additionally, it has been numerically shown that using state-action-dependent baselines does not reduce variance compared to state-dependent

baselines in benchmark environments (Tucker et al., 2018).

**Implementation matters in deep policy gradients.** Despite its accomplishments, deep RL methods are notorious for their brittleness, lack of stability and reliability (Henderson et al., 2018; Engstrom et al., 2020). Furthermore, it has been reported that the performance of PPO heavily relies on code-level optimization techniques, due to its sensitivity to hyperparameter choices (Henderson et al., 2018; Engstrom et al., 2020) and regularization techniques (Liu et al., 2021). Recent works aimed at systematically understanding and addressing hyperparameter tuning issues in deep RL have been developed (Paul et al., 2019; Eimer et al., 2023; Adkins et al., 2024). We contribute to this line of research by directly identifying value estimation as the core component that makes deep policy gradients work.

## 3. Preliminaries

Consider an infinite-horizon Markov decision process (MDP), defined by the tuple $(\mathcal{S}, \mathcal{A}, \Phi, r, \rho_0, \gamma)$, where the state space $\mathcal{S}$ and the action space $\mathcal{A}$ are continuous and compact, $\Phi : \mathcal{S} \times \mathcal{S} \times \mathcal{A} \to [0, \infty)$ is the transition probability density function, $R : \mathcal{S} \to \mathbb{R}$ is the reward function, $\rho_0$ is the distribution of the initial state $s_0$, and $\gamma \in (0, 1)$ is the discount factor. For a parameterized stochastic policy $\pi_\theta$, the policy objective to maximize is given by

$$J(\theta) = \hat{\mathbb{E}}_{(s_t, a_t) \sim \pi_\theta, s_0 \in \rho_0} \Big[ \sum_{t=0}^{\infty} \gamma^t R(s_t, a_t) \Big], \quad (1)$$

where $\theta \in \mathbb{R}^N$ is the policy parameter. According to (Sutton et al., 1999), the original form of policy gradient estimator is given as:

$$\nabla_\theta J(\theta) \propto \int_{\mathcal{S}} \rho^\pi(s) \int_{\mathcal{A}} Q^\pi(s, a) \nabla_\theta \pi_\theta(a|s) \, \mathrm{d}a \mathrm{d}s, \quad (2)$$

where $Q^\pi$ is the $Q$-function of the current policy $\pi$ and $\rho^\pi$ the discounted visitation frequencies. While there are various methods for approximating $Q$ values, the most commonly used approach is Monte-Carlo, which estimates the discounted return for each trajectory, i.e., $\hat{G}_t = \sum_{k=t}^{T} \gamma^{k-t} R_k$. The vanilla policy gradient loss is given as

$$L(\theta) = \hat{\mathbb{E}}_{(s_t, a_t) \sim \pi_\theta} \Big[ \log \pi_\theta(a_t|s_t) \, \hat{G}_t \Big], \quad (3)$$

which is the REINFORCE algorithm (Williams, 1992).

**REINFORCE with baseline.** In practice, the variance of the estimation $\hat{Q}_t$ can be large, making it difficult to distinguish between higher-valued actions and less highly valued ones. This motivates the use of baseline functions to reduce the variance (Sutton & Barto, 1998). In this case,

the objective is given by:

$$L^{PG}(\theta) = \hat{\mathbb{E}}_{(s_t, a_t) \sim \pi_\theta} \Big[ \log \pi_\theta(a_t|s_t) \, \hat{A}_t \Big], \qquad (4)$$

where $\hat{A}_t = \hat{Q}_t - \hat{V}(s_t)$ is the estimated advantage of $(s_t, a_t)$, and $\hat{V}$ is the estimated value function of the current policy $\pi$ which serves as the baseline. Let $V_\phi(s) = V(s; \phi)$ denote the value approximation to the true value function $V^{\pi_\theta}$ by a neural network where $\phi \in \mathbb{R}^M$ is the network parameter.

In some literature, this algorithm is also referred to as Advantage Actor-critic (A2C) (Mnih et al., 2016)). To avoid any potential ambiguity, we henceforth refer to the objective in equation (4) whenever mentioning the vanilla policy gradient (VPG) algorithm. In the following sections, we will demonstrate that optimizing the objective in equation (4) is sufficient to achieve performance comparable to PPO, provided that the baseline function $\hat{V}$ is properly optimized.

**Trust region methods.** According to the conservative policy iteration method (Kakade & Langford, 2002), a small policy update that improves the corresponding surrogate objective is guaranteed to improve the true policy objective. This result motivated the TRPO algorithm (Schulman et al., 2015a), which optimizes the following surrogate objective:

$$L^{TRPO}(\theta) = \hat{\mathbb{E}}_{(s_t, a_t) \sim \pi_\theta} \Big[ \frac{\pi_\theta(a_t|s_t)}{\pi(a_t|s_t)} \hat{A}_t \Big], \qquad (5)$$

which is further subject to a constraint on the KL divergence between the current and the old policy:

$$\hat{\mathbb{E}}_{s \sim \pi} \Big[ D_{KL}(\pi_\theta(\cdot|s) \,\|\, \pi(\cdot|s)) \Big] \leq \delta, \qquad (6)$$

for some $\delta > 0$. In practice, however, TRPO can be expensive, as it involves the computation of second-order derivatives. This motivates the PPO algorithm (Schulman et al., 2017), which simplifies the objective and avoids the need for second-order derivatives:

$$L^{PPO}(\theta) = \hat{\mathbb{E}}_\pi \Big[ \min \Big( r_t \hat{A}_t, \text{clip}(r_t, 1 - \epsilon, 1 + \epsilon) \hat{A}_t \Big) \Big], \tag{7}$$

where $r_t = \frac{\pi_\theta}{\pi}$ is the probability ratio from importance sampling, and $\epsilon$ is the clipping parameter.

## 4. Revisiting Trust Regions

According to the theoretical analysis in TRPO (Schulman et al., 2015a), the true performance $J(\cdot)$ is guaranteed to improve when a policy update is made to improve the TRPO surrogate $L^{TRPO}(\cdot)$, provided that the KL divergence between two consecutive policies is small enough. This condition is expressed through the following inequality:

$$J(\theta) \geq L^{TRPO}(\theta) - \frac{4\beta\gamma}{(1 - \gamma)^2} D_{KL}^{max}(\pi_\theta \,\|\, \pi_{\theta_{old}}) \quad (8)$$

where $\beta = \max_{s,a} |A_\pi(s, a)|$ is the maximum advantage value. In practice, the TRPO algorithm solves the following approximate constrained optimization problem (Achiam, 2018):

$$\underset{\theta}{\text{maximize}} \; \nabla_\theta L^{TRPO}(\theta) \Big|_{\theta = \theta_{old}} \cdot (\theta - \theta_{old})$$

$$\text{subject to} \; \frac{1}{2}(\theta - \theta_{old})^T H(\theta - \theta_{old}) \leq \delta$$

where $H$ is the Hessian of $\hat{\mathbb{E}}_{s \sim \pi} \Big[ D_{KL}(\pi_\theta(\cdot|s)|\pi(\cdot|s)) \Big]$, which provides a quadratic approximation to the KL constraint. Solving with conjugate gradient methods leading to the update rule $\theta_{k+1} = \theta_k + \alpha^j \sqrt{\frac{2\delta}{g^T H g}} H^{-1} g$, where $g = \nabla_\theta L^{TRPO}(\theta)|_{\theta = \theta_k}$ denotes the gradient at $\theta = \theta_k$, $\alpha \in (0, 1)$ is the backtracking coefficient, and $j$ is the smallest nonnegative integer that makes the update satisfy the KL constraint (6) while producing a positive surrogate advantage.

However, there are two issues with the design in practice. First, although it is proven that $\nabla_\theta L^{TRPO}(\theta)|_{\theta = \theta_k} = \nabla_\theta J(\theta)|_{\theta = \theta_k}$ under conditions that *assume* the existence of gradients (Schulman et al., 2015a), the gradient $\nabla_\theta J(\theta)$ may not exist for many continuous control problems in the first place. This is because the policy optimization landscapes in these cases often exhibit fractal structures, as illustrated in Figure 1 (Wang et al., 2023). Consequently, the linear approximation used in the TRPO implementation may poorly represent the original objective, making it difficult to attribute the success of TRPO to this approximation.

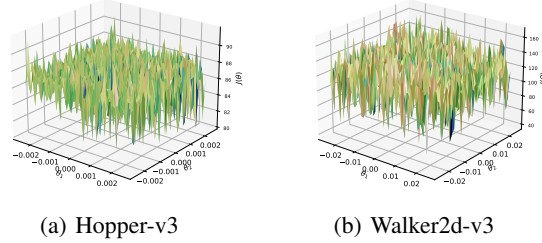

(a) Hopper-v3        (b) Walker2d-v3

*Figure 1.* Policy objectives in many continuous-control environments are highly non-smooth and fractal.

Second, since the approximate objective is linear in $\theta$ and the quadratic constraint induces a convex feasible region, the solution to this problem should always lie on the boundary of the feasible region. As a result, the practical algorithm attempts to take the *largest* possible step in each iteration, whereas the theory suggests taking the *smallest* step to guarantee policy improvement. This discrepancy suggests that the effectiveness of trust region methods may not stem solely from their enforcement of theoretical constraints, but rather from hyperparameter tuning and practical

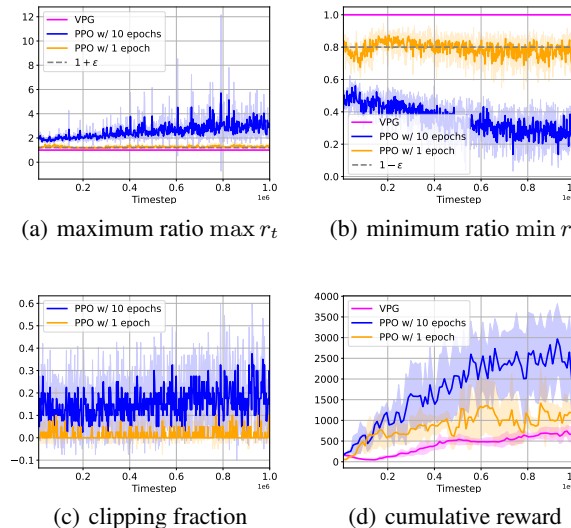

(a) maximum ratio $\max r_t$  (b) minimum ratio $\min r_t$

(c) clipping fraction  (d) cumulative reward

*Figure 2.* We compare the performance of different implementations on the MuJoCo Hopper task. The clipping parameter is set to $\epsilon = 0.2$ as default.

code-level optimizations that extend beyond the theoretical framework (Henderson et al., 2018).

**Ratio trust regions in PPO.**   To simplify implementation, PPO instead defines trust regions using the importance sampling ratio rather than KL divergence. However, despite the potential correlation, there is no strong correspondence between enforcing a trust region and policy improvement. Notably, during PPO training, the maximum probability ratio $\max_t r_t$ can significantly violate the trust region defined by the clipping coefficient $\epsilon$ (Engstrom et al., 2020). This occurs because the gradient $\nabla_\theta r_t$ vanishes outside the trust region (e.g., when $r_t > 1 + \epsilon$) due to the clipping operator. Consequently, if an excessively large step is taken on $\pi_\theta(a_t|s_t)$ such that $r_t$ leaves the trust region, it may not be pulled back to the trust region anymore due to the vanishing gradient.

Our empirical results support this observation. Figure 2 (a)-(c) show that the standard PPO implementation with 10 epochs per iteration, despite employing the ratio clipping mechanism, consistently violates the ratio bound, coinciding with findings from prior work. In contrast, PPO with only 1 epoch per iteration and VPG both successfully enforce the trust region defined by the ratio bound. However, they are outperformed by the standard PPO implementation as shown in Figure 2 (d).

# 5. Importance of Value Estimation

In the previous section, we demonstrated that there is no direct relationship between enforcing a trust region and im-

proving algorithmic performance. Nevertheless, trust region methods generally outperform VPG, so it remains to identify the fundamental factors that drive policy improvement in practice. Given that the number of optimization steps plays a crucial role in the outcome, we focus on analyzing this aspect throughout this section and argue that value estimation is the core factor driving policy improvement.

### 5.1. Increased value steps in TRPO

The VPG algorithm employs a single optimization loop to simultaneously optimize the policy and value networks using automatic differentiation software. In this loop, the value network is trained to minimize the regression loss:

$$L^V(\phi) = \|V_\phi - \hat{V}_{target}\|^2_{\mathcal{D}} \tag{9}$$

where $\mathcal{D}$ represents the collected data, $V_\phi$ is the parameterized value approximation, and $\hat{V}_{target}$ is the target value, typically obtained through Temporal Difference (TD) estimates. For instance, the target value is given by $\hat{V}_{target}(s_t) = \sum_{k=t}^{T} \gamma^{k-t} R_k + \gamma^{T+1-t} V_\phi(s_{T+1})$ when the GAE factor $\lambda = 1$.

However, the TRPO implementation requires separate optimization loops for the policy and value networks: the policy is optimized using the conjugate gradient algorithm to enforce the trust region constraint, while the value network is updated similarly to VPG. Perhaps surprisingly, this seemingly small modification in the code-level implementation may account for the fundamental difference between the two algorithms. Specifically, the VPG algorithm performs only one optimization step per epoch, which is insufficient for the value network to accurately estimate the true returns. In contrast, TRPO applies multiple optimization steps per iteration for the value network without over-optimizing the policy network. This results in more accurate value estimation and, consequently, better performance. Some examples are presented in Table 1.

| RL libraries | # Value steps | Learning rate |
|---|---|---|
| OpenAI Baselines | 5 | 0.001 |
| Spinning Up | 80 | 0.001 |
| Stable-Baseline3 | 10 | 0.001 |
| Tianshou | 20 | 0.001 |

*Table 1.* The number of value steps per iteration and the learning rate for value networks in default settings of several deep RL libraries (Dhariwal et al., 2017; Achiam, 2018; Raffin et al., 2021; Weng et al., 2022).

### 5.2. How PPO addresses value estimation?

Having a properly optimized value network is also the key reason why PPO outperforms VPG. To understand this, we analyze two algorithmic techniques in the PPO algorithm

and examine how they contribute to improved value estimation.

**Mini-batching and multiple epochs.** In practice, both of them contribute to increasing the number of value steps in each iteration, as we have

$$\text{\# gradient steps} = \text{\# epochs} \times \frac{\text{full-batch size}}{\text{mini-batch size}}$$

Therefore, if the entire batch is used with the same number of epochs, the value network will be under-approximated, leading to inaccurate baseline estimation. As shown in Figure 3 (a), PPO completely fails to find a good policy when the entire batch is used for policy training. This failure can be attributed to the poor value estimation displayed in Figure 3 (b), where the value estimation error is calculated through

$$\eta(s_0) = \sum_{k=0}^{T} \gamma^{k-t} R(s_t, a_t) - V(s_0; \phi) \qquad (10)$$

where $a_t \sim \pi_\theta$ and $s_0 \sim \rho_0$. This also helps explain why PPO with a single epoch performs poorly as seen in Figure 2.

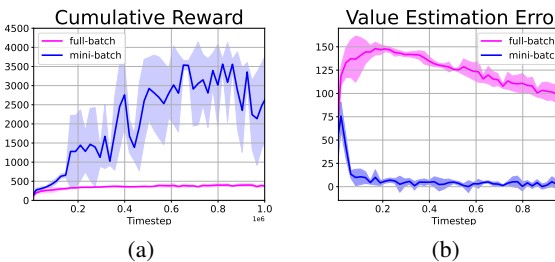

(a)                (b)

*Figure 3.* The cumulative reward and value estimation error during PPO training in the Hopper task are compared between full-batch and mini-batch updates. It highlights how the use of full-batch updates leads to suboptimal policy performance, as reflected in the large value estimation errors, while mini-batch updates facilitate more accurate value estimation and better cumulative reward outcomes.

**Probability ratio clipping.** Like VPG, PPO uses a single optimization loop for both networks as well. Therefore, while increasing the number of optimization steps improves value estimation, it may also lead to over-updating of the policy network. The clipping mechanism in PPO mitigates this by blocking excessive updates made to the policy network, allowing the value network to catch up with true returns.

### 5.3. Theoretical analysis

We now show that this value estimation issue can be explained from the perspective of optimization landscapes in the policy space. Briefly speaking, the policy objective

changes rapidly in many continuous-control environments, which necessitates more update steps for the value network in each iteration. In dynamical systems theory, maximal Lyapunov exponents are used to study chaotic behaviors: given a dynamical system $s_{t+1} = F(s_t), s_0 \in \mathbb{R}^n$, and a small perturbation $\Delta Z_0$ made to $s_0$, the divergence caused at time $t$ is denoted by $\Delta Z(t)$. For chaotic systems, their dynamics are sensitive to initial conditions so that it has

$$\|\Delta Z(t)\| \simeq e^{\lambda t} \|\Delta Z_0\|$$

for some $\lambda > 0$. The rigorous definition of Maximal Lyapunov exponents is presented below:

**Definition 5.1.** (Maximal Lyapunov exponent (Lorenz, 1995)) For the dynamical system $s_{t+1} = F(s_t), s_0 \in \mathbb{R}^n$, the maximal Lyapunov exponent $\lambda_{\max}$ at $s_0$ is defined as the largest value such that

$$\lambda_{\max} = \limsup_{t \to \infty} \limsup_{\|\Delta Z_0\| \to 0} \frac{1}{t} \log \frac{\|\Delta Z(t)\|}{\|\Delta Z_0\|}. \qquad (11)$$

Specifically, according to Proposition C.2, we have the following estimation for the policy objective update:

$$|J(\theta') - J(\theta)| \sim \mathcal{O}(\|\theta' - \theta\|^\alpha) \qquad (12)$$

where $\alpha = \frac{-\log \gamma}{\lambda(\theta)}$ and $\lambda(\theta)$ is the maximal Lyapunov exponent of the dynamics controlled by $\pi_\theta$, which is typically positive in many continuous-control environments. Thus, we may have $\alpha < 1$ when $\lambda(\theta) > -\log \gamma$, which is generally the case when the underlying dynamics is chaotic (Lorenz, 1995). This results in fractal landscapes in the policy space as illustrated in Figure 1, and a rapidly changing policy objective.

The above estimation also suggests that even a small update in the parameters can lead to a significant change in the value function of the policy objective $J(\theta)$. On the other hand, the training of value networks is generally more stable. For most activation functions used in neural networks, such as $\tanh$ and ReLU, the network output is Lipschitz continuous with respect to both the input variables and the network parameters. Specifically, for a given state $s$ and parameters $\phi$ and $\phi'$, the difference in the value network output can be estimated by

$$|V(s; \phi') - V(s; \phi)| \sim \mathcal{O}(\|\phi' - \phi\|) \qquad (13)$$

when $\|\phi' - \phi\|$ is sufficiently small. This suggests that the landscape of the value regression loss in equation (9) is smooth (e.g., as illustrated in Figure 12), such that small updates in the parameters always lead to small changes in the estimated values. We summarize this analysis with the following theorem, which provides an estimate of the relationship between the updating rates of the value and policy networks:

**Theorem 5.2.** *Assume that the dynamics, reward function, policy and value networks are all Lipschitz continuous with respect to their input variables. Let $\beta_1, \beta_2$ denote the learning rate for policy and value network, respectively, and $K_V$ denote the number of value steps per epoch. Then for each policy step, a good value estimation requires*

$$K_V \geq \frac{K_1 \beta_1^\alpha}{K_2 \beta_2}$$

*steps made to the value network when $\beta_1, \beta_2$ are small. $\alpha = \frac{-\log \gamma}{\lambda(\theta)} < 1$ and $\lambda(\theta)$ is the maximal Lyapunov exponent of the dynamics controlled by $\pi_\theta$, $\gamma \in (0,1)$ the discount factor and $K_1, K_2 > 0$ are constant independent of the learning rates $\beta_1$ and $\beta_2$.*

This result suggests that the value network should be optimized more aggressively than the policy network, with more optimization steps and/or higher learning rates, to ensure that the value estimates made by the critic network can closely track the true values observed in rollouts. Furthermore, to prevent interference between the policy and value networks, separate networks should be used, as also suggested in (Cobbe et al., 2021; Huang et al., 2022a). Additional details can be found in Appendix C.

## 6. Experiments

In the previous section, we theoretically demonstrated that value estimation is the core component of on-policy algorithms, and by optimizing it, a simple method is expected to perform as well as more advanced methods. In this section, we evaluate the role of value estimation across a range of continuous-control tasks from the OpenAI Gym benchmarks (Brockman et al., 2016). Specifically, we first show that the VPG algorithm, with a properly optimized value network, can achieve performance comparable to or even better than PPO with their corresponding default settings. Additionally, we conduct ablation studies on several code-level optimizations that may impact the performance of value estimation in both VPG and PPO. The experimental setup is detailed in Appendix A, and further experimental results can be found in Appendix B.

We adapt the implementation and PPO baseline from Tianshou (Weng et al., 2022). We also provide a single-file codebase modified from CleanRL (Huang et al., 2022b) which enables implementations in other environments such as DeepMind Control (Tunyasuvunakool et al., 2020) and Isaac Gym (Makoviychuk et al., 2021). Code for the empirical results is available at https://github.com/taowang0/value-estimation-vpg.

**Vanilla policy gradient with multiple value steps.** The performance of the VPG algorithm with different numbers of value steps is compared to PPO in Figure 4. The results show that by simply increasing the number of value steps, VPG can eventually achieve performance comparable to PPO in the Halfcheetah, Hopper, and Walker environments, and outperform PPO in the higher-dimensional Ant and Humanoid environments. The performance of VPG stabilizes when the number of value steps reaches 50, at which point the value network is able to accurately capture the true returns, leading to more precise advantage estimations. Full comparisons of algorithmic performance and corresponding value estimation errors for different implementations are presented in Figure 4 and Figure 5. Additionally, Figure 10 shows that VPG is more efficient than PPO in terms of policy steps. Also, VPG consistently improves the policy across all environments, while PPO's performance drops after 2M environment steps in the Walker and Ant tasks. Each task is trained with five random seeds, with solid curves representing the average return and shaded regions indicating the one standard deviation confidence interval.

**GAE factor $\lambda$.** The effect of the Generalized Advantage Estimation (GAE) factor is to reduce the variance of value estimation in on-policy algorithms (Schulman et al., 2015b). To assess how the GAE factor influences the performance of VPG and PPO, we consider two cases: $\lambda = 0.95$ (default GAE factor) and $\lambda = 1$ (equivalent to Monte-Carlo). As shown in Figure 6, both the reward curves of VPG and PPO drop when $\lambda = 1$ compared to $\lambda = 0.95$, which can be attributed to the variance reduction effect of the GAE method. Notably, while the performance of VPG with 100 value steps is only slightly affected, the change in the GAE factor has a significant impact on PPO, as shown in Table 2. For example, in Walker and HalfCheetah, PPO's cumulative reward drops by more than 40% when $\lambda = 1$.

**Learning rate of policy network.** In general, the sensitivity to learning rate in deep RL is at a similar level to that in supervised learning. Most commonly-used learning rates from 0.0003 to 0.001 work well for VPG. One caveat is that excessively small learning rates do not always guarantee policy improvement due to the fractal landscapes in value and policy space. On the other hand, using learning rates that are too large leads to poor performance, too. The reason is that unlike in supervised learning where the loss is optimized for multiple steps which corrects the potentially wrong direction generated in the first step, VPG only performs one policy update per collected batch of data and thus has no chance to subsequently correct the error. In Figure 6, we can observe that PPO suffers from the sensitivity to learning rates in the Ant environment, while the performance of VPG remains relatively stable across different learning rates.

**Reward normalization.** While increasing the number of value steps is the most straightforward way to improve value estimation, reward normalization can impact the effective-

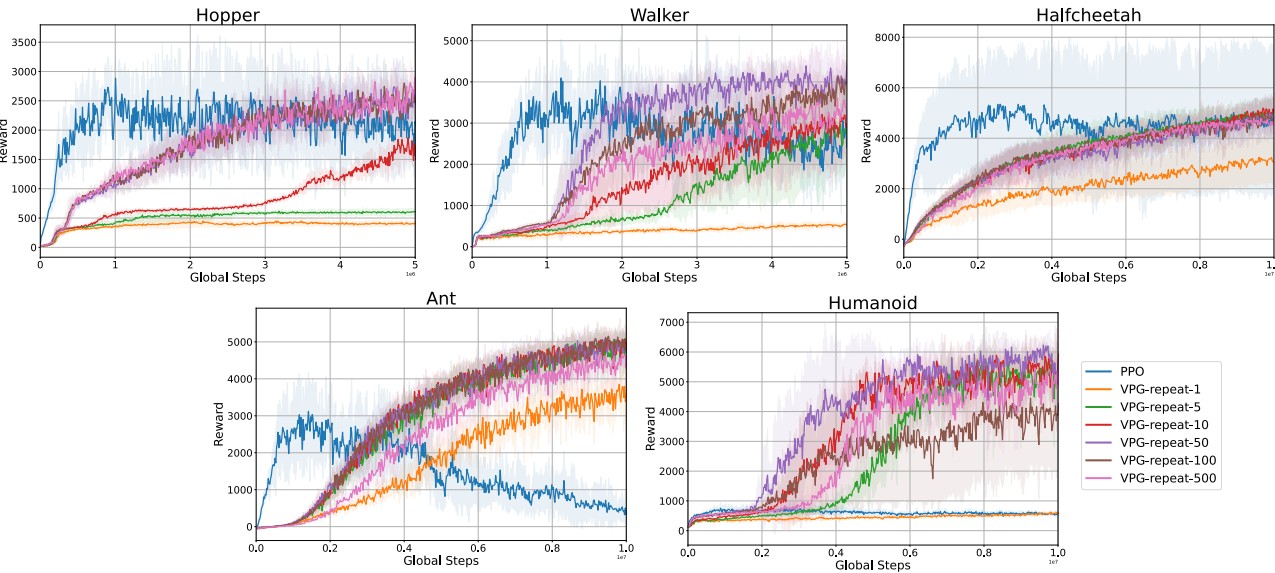

*Figure 4.* Training curves on Gymnasium benchmarks. The curve *VPG-repeat-k* corresponds to the vanilla policy gradient algorithm with $k$ value steps applied each iteration. For example, *VPG-repeat-1* represents the original vanilla policy gradient implementation. As the number of value steps increases, the performance of vanilla policy gradient consistently improves, eventually converging to or outperforming PPO when the number of value steps reaches 50 or more.

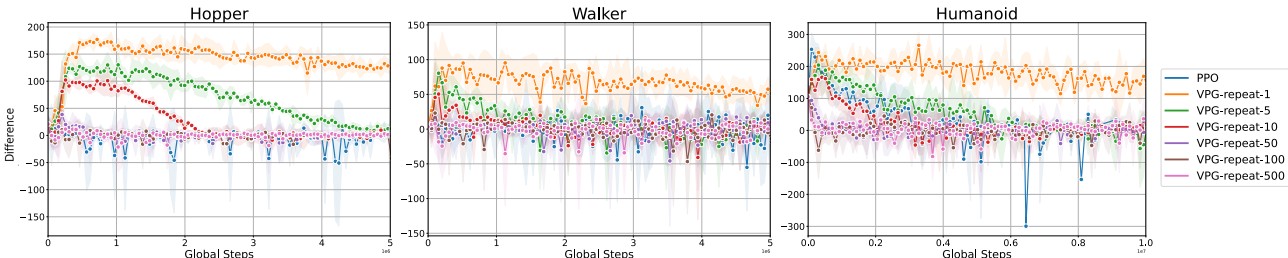

*Figure 5.* The corresponding value estimation difference in the experiments shown in Figure 10. We disabled exploration during evaluation, using the deterministic policy as the direct output of the policy network. The difference in value estimation is computed through Equation 10. We observe a clear correlation between the value estimation difference in VPG and its performance. As the value steps increase, the estimation error decreases and eventually oscillates around zero, leading to improved performance. More results can be found in Figure 11.

ness of value estimation as well. The PPO algorithm employs a reward normalization scheme that rescales the rewards in the current batch by dividing them by the standard deviation of a rolling discounted sum, without altering the mean. As shown in Figure 9, reward scaling improves the performance of VPG when the number of value steps is 1 or 10, but has no significant effect when the number of value steps is 100. This suggests that reward normalization helps reduce the magnitude of rewards when it is large, thereby decreasing the error in value estimation. However, its effect is negligible when the value network is already properly optimized, as in the case of VPG with 100 value steps per iteration and PPO.

**Mini-batching and multiple epochs.** In Figure 2, we observed that using full batches in PPO can lead to poor value estimation due to an insufficient number of value steps. Figure 9 (b) illustrates how varying mini-batch sizes significantly affects PPO's performance in the Hopper task. Additionally, the learning curves in Figure 10 show that the default hyperparameters are not optimal for the Ant and Humanoid tasks, where the dynamics are more complex, making the policy gradient estimator effective only for small policy updates. As shown in Figure 9, PPO performs significantly better with larger mini-batches and fewer policy epochs on these two tasks, both of which reduce the number of policy updates per iteration.

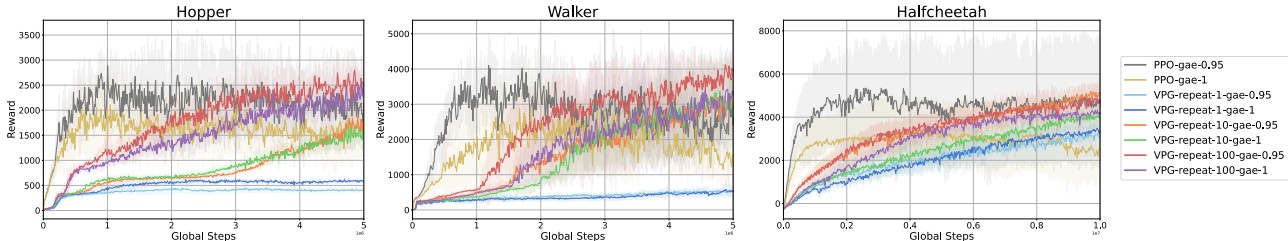

*Figure 6.* The influence of GAE factor $\lambda$ across three tasks.

| Algorithm | GAE $\lambda$ | Hopper | Walker | HalfCheetah |
|---|---|---|---|---|
| VPG | 0.95 | $2601.57 \pm 232.14$ | $3457.79 \pm 646.70$ | $4928.88 \pm 807.04$ |
| VPG | 1 | $2323.28 \pm 436.24$ | $3123.83 \pm 987.58$ | $4381.30 \pm 222.15$ |
| PPO | 0.95 | $1965.29 \pm 478.14$ | $2527.74 \pm 507.40$ | $4488.60 \pm 2699.54$ |
| PPO | 1 | $1611.46 \pm 541.32$ | $1431.50 \pm 612.17$ | $2604.08 \pm 1237.80$ |

*Table 2.* Performance of VPG and PPO with different GAE factors after 5M environment steps for Hopper and Walker, 10M environment steps for HalfCheetah.

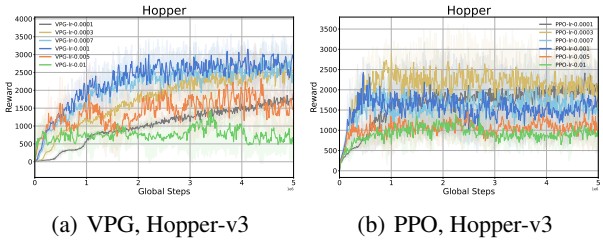

(a) VPG, Hopper-v3      (b) PPO, Hopper-v3

*Figure 7.* The influence of learning rates in VPG and PPO.

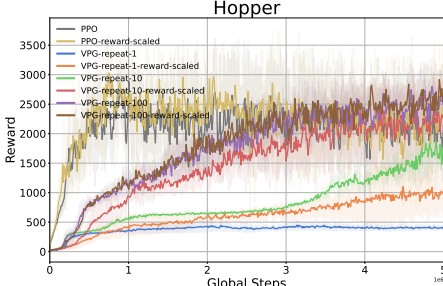

*Figure 8.* The influence of reward normalization in different algorithms. We see that although normalizing rewards can improve the performance of VPG when it has 1 and 10 value steps, it does not make significant difference to VPG when there are as many as 100 value steps per iteration.

# 7. Conclusion

In this work, we provide both theoretical and empirical analyses to demonstrate that the performance of a policy gradient algorithm is primarily determined by how accurately the value function is estimated, rather than by any specific policy objective design adopted for policy training. Our findings suggest that deep on-policy algorithms may work for a very simple reason that even the simplest vanilla policy gradient method can accomplish.

In the meantime, we should clarify that the ratio clipping mechanism in PPO, although it does not fundamentally contribute to performance improvement, still has its strengths in that it allows multiple policy updates on a single batch, thereby enhancing sample efficiency, sometimes at the cost of reduced robustness. This point is evident in the Ant and Humanoid tasks, where the default PPO implementation updates the policy too aggressively and leads to suboptimal performances. Although our argument may, in principle, be generalizable to other domains such as large language models (LLMs), the present analysis is confined to continuous control problems, and we refrain from asserting its validity

beyond this specific context.

There are two key messages we want to convey through this work: First, value estimation is perhaps the most crucial component of on-policy algorithms. By optimizing it, we can significantly improve their performance. This finding encourages us to revisit the foundations of the development of policy gradient methods (namely, from VPG to TRPO/PPO), as the core ideas in these methods (e.g., enforcing a trust region, performing multiple policy updates) may not be the fundamental underpinnings of the improved performance observed in practice. Second, our results indicate that it is possible to develop on-policy algorithms that are robust without the extensive need for hyperparameter tuning and code-level optimization techniques. For instance, VPG with multiple value steps performs well and requires very few hyperparameters to tune, suggesting that complex

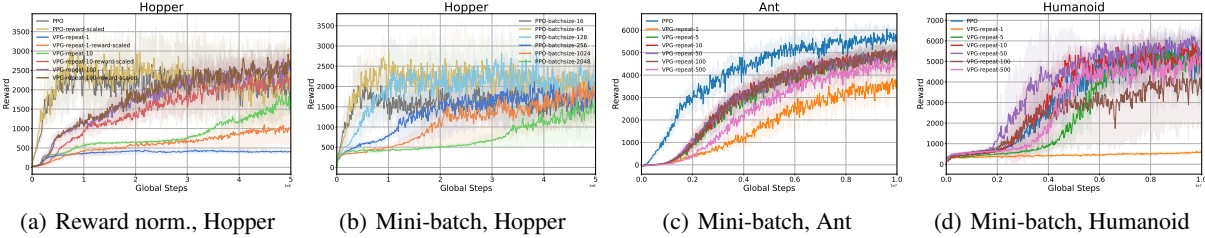

(a) Reward norm., Hopper  (b) Mini-batch, Hopper  (c) Mini-batch, Ant  (d) Mini-batch, Humanoid

*Figure 9.* Code-level optimizations in policy gradient methods. (a) Although normalization rewards can improve the performance of VPG when *# value steps* = 1, its strength is rather limited compared to the fully-optimized VPG with *# value steps* = 100. (b) The performance of PPO varies with different mini-batch sizes where the number of epochs is fixed at 10. (c) (d) We use *mini-batch size = 512, # epoch = 2* for PPO in Ant and Humanoid which significantly outperform the default implementation in Figure 4.

algorithmic architectures like PPO may not be necessary for effective policy gradient methods. This is important for deep RL in real-world applications, especially for practitioners who want to leverage policy gradient methods in their domains but have little or no expertise in RL. With well-developed parallel computing architectures that accelerate simulation and data collection, vanilla policy gradient has the potential to serve as a robust and effective alternative to PPO in various robot learning tasks.

## Acknowledgements

The authors would like to thank Yuexin Bian for her valuable suggestions in developing the codebase, and the anonymous reviewers for their helpful comments in revising the paper. This material is based on work supported by NSF Career CCF 2047034, NSF CCF DASS 2217723, and NSF AI Institute CCF 2112665.

## Impact Statement

This paper presents work whose goal is to advance the field of Machine Learning. There are many potential societal consequences of our work, none which we feel must be specifically highlighted here.

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

## A. Experiment Hyperparameters

|                                       | PPO      | VPG      |
|---------------------------------------|----------|----------|
| Num. env.                             | 16       | 16       |
| Discount factor ($\gamma$)            | 0.99     | 0.99     |
| Num. epochs                           | 10       | 1        |
| Batch size                            | 2048     | 2048     |
| Minibatch size                        | 64       | 2048     |
| GAE factor ($\lambda$)                | 0.95     | 0.95     |
| Optimizer                             | Adam     | Adam     |
| Clipping parameter ($\epsilon$)       | 0.2      | N/A      |
| Advantage normalization               | False    | False    |
| Observation normalization             | True     | True     |
| Reward normalization                  | False    | False    |
| Learning rate decay                   | False    | False    |
| Entropy coefficient                   | 0        | 0        |
| Policy network                        | [64, 64] | [64, 64] |
| Value network                         | [64, 64] | [64, 64] |
| Activation function                   | tanh     | tanh     |
| Gradient clipping ($l_2$ norm)        | 1.0      | 1.0      |

*Table 3.* Default hyperparameters for policy gradient algorithms.

|     | Hopper | Walker | HalfCheetah | Ant    | Humanoid |
|-----|--------|--------|-------------|--------|----------|
| VPG | 0.0003 | 0.0007 | 0.0007      | 0.0007 | 0.0007   |
| PPO | 0.0003 | 0.0003 | 0.0003      | 0.0003 | 0.0003   |

*Table 4.* The learning rate used for the policy and value network in each task. Note that the learning rate 0.0003 is specifically chosen for Hopper to allow for a better comparison with PPO on that task. As shown in Figure 7 (a), using the learning rate 0.0007 in Hopper actually results in better performance.

# B. Additional Experimental Results

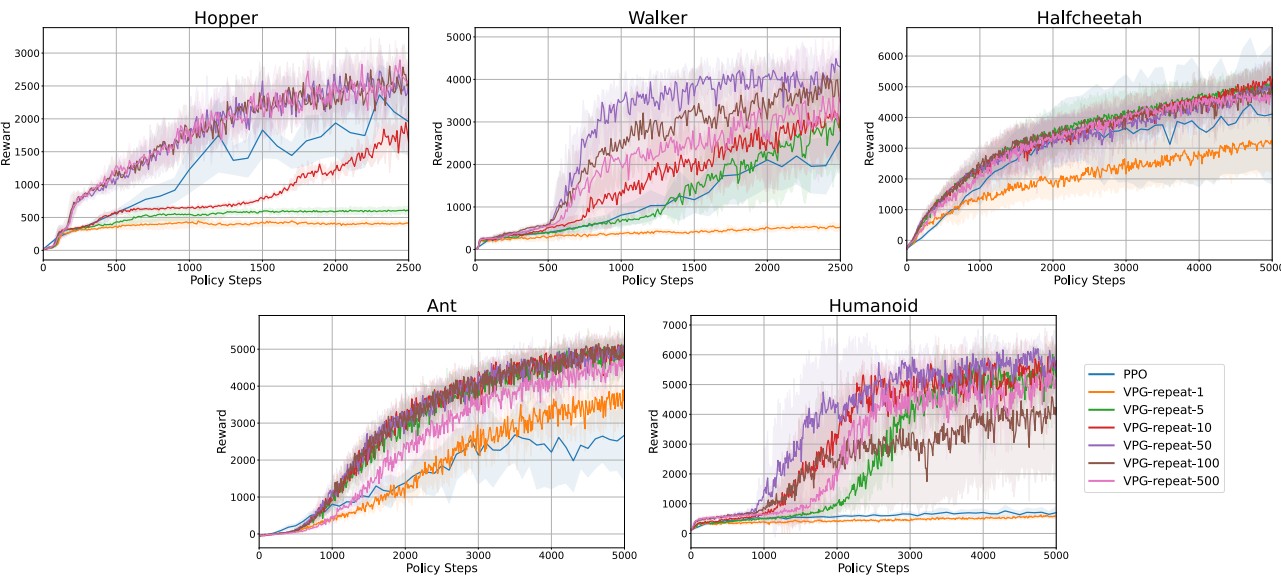

*Figure 10.* The cumulative rewards versus policy step in each MuJoCo task.

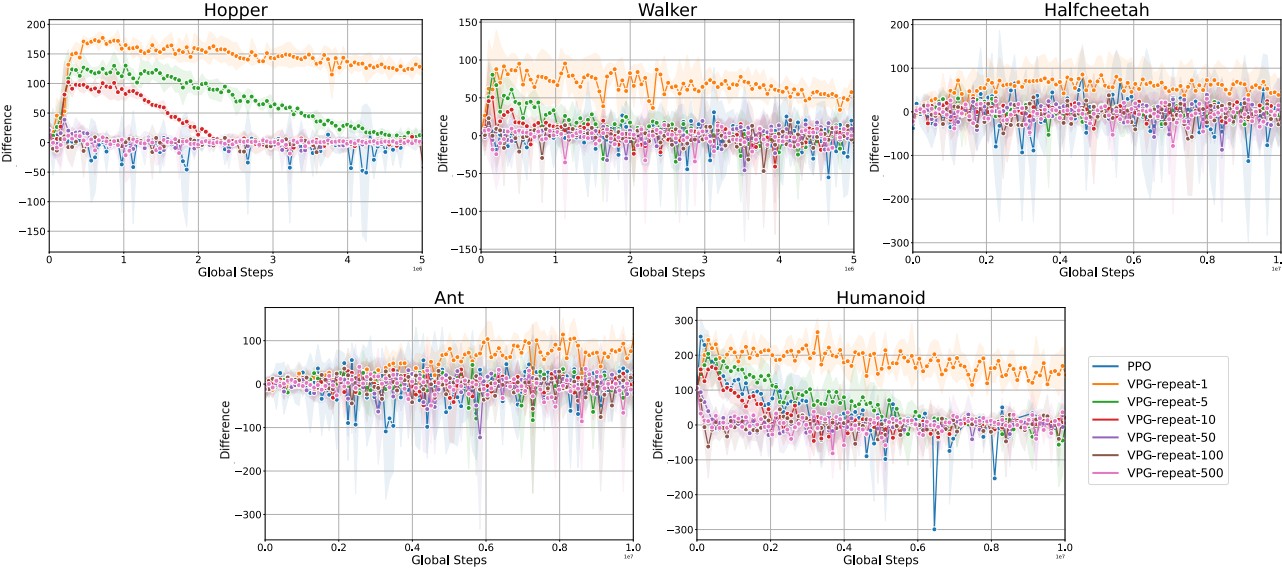

*Figure 11.* A full comparison of value estimation difference for each implementation.

## C. Theoretical Analysis of Value Estimation

Existing work has shown that the optimization landscape in the policy space is fractal in many continuous-control environments. To better characterize this behavior, we first introduce the concept of Hölder continuity:

**Definition C.1.** (Hölder continuity) Let $f : \mathbb{R}^{k_1} \to \mathbb{R}^{k_2}$ be a function. Given $x \in \mathbb{R}^{k_1}$ and $\alpha \in (0, 1]$, we say that $f$ is $\alpha$-Hölder continuous at $x$ if for any $\delta > 0$, there exists $C_1 > 0$ such that

$$\|f(x') - f(x)\| \leq C_1 \|x' - x\|^\alpha$$

for all $x' \in \mathbb{R}^{k_1}$ that has $\|x' - x\| \leq \delta$.

Note that Hölder continuity is equivalent to Lipschitz continuity when $\alpha = 1$, and the function can be highly non-smooth and fractal when $\alpha < 1$. The following theorem establishes a connection between the chaotic behavior in an MDP and the smoothness of the corresponding policy optimization objective:

**Proposition C.2.** *(Wang et al., 2023) Assume that the dynamics, reward function and policy are all Lipschitz continuous with respect to their input variables. Let $\pi_\theta$ be a deterministic policy and $\lambda(\theta)$ denote the maximal Lyapunov exponent of the dynamics. Suppose that $\lambda(\theta) > -\log\gamma$ and let $\alpha = \frac{-\log\gamma}{\lambda(\theta)}$, then*

1. *Value function $V^{\pi_\theta}(s)$ is $\alpha$-Hölder continuous in the state $s \in \mathcal{S}$;*

2. *Q-funtion $Q^{\pi_\theta}(s, a)$ is $\alpha$-Hölder continuous in the action $a \in \mathcal{A}$;*

3. *Policy objective $J(\theta)$ is $\alpha$-Hölder continuous in the policy parameter $\theta \in \mathbb{R}^N$.*

**Proof of Theorem 5.2.** According to Proposition C.2, we have the following estimation:

$$|J(\theta') - J(\theta)| \sim \mathcal{O}(\|\theta' - \theta\|^{\frac{-\log\gamma}{\lambda(\theta)}}) \tag{14}$$

where $\lambda(\theta)$ is the maximal Lyapunov exponent of the dynamics controlled by $\pi_\theta$, which is typically positive in many continuous-control environments. Illustrations of fractal landscapes in several MuJoCo environments are shown in Figure 1, where even a small update in the parameters can lead to a significant change in the value of the policy objective $J(\theta)$.

Unlike policy networks, which can lead to dramatic changes in the return even with a single parameter update, the training of value networks is generally more stable, as it minimizes the regression loss:

$$L^V(\phi) = \|V_\phi - \hat{V}_{target}\|_{\mathcal{D}}^2 \tag{15}$$

where $\mathcal{D}$ represents the collected data, $V_\phi$ is the parameterized value approximation, and $\hat{V}_{target}$ is the target value.

For most activation functions used in neural networks, such as $\tanh$ and ReLU, the network output is Lipschitz continuous with respect to both the input variables and the network parameters. Specifically, for a given state $s$ and parameters $\phi$ and $\phi'$, the difference in the value network output can be estimated by

$$|V(s; \phi') - V(s; \phi)| \sim \mathcal{O}(\|\phi' - \phi\|) \tag{16}$$

when $\|\phi' - \phi\|$ is sufficiently small. This suggests that the landscape of the value regression loss in equation (9) is smooth (e.g., as illustrated in Figure 12), such that small updates in the parameters always lead to small changes in the estimated values.

Now, let us consider the policy improvement. Let $\theta_k$ and $\phi_k$ denote the parameters of the policy and value networks at the $k$-th iteration, respectively. Note that equation (14) suggests that for a fixed state $s$, the discounted return $\sum_{t=0}^{\infty} \gamma^t R_t$ is $\frac{-\log\gamma}{\lambda(\theta)}$-Hölder continuous with respect to the policy parameter. Therefore, the difference in the target value between two consecutive policy steps can be estimated by

$$|V_{target,k+1}(s) - V_{target,k}(s)| \simeq K_1 \beta_1^{\frac{-\log\gamma}{\lambda(\theta_k)}}$$

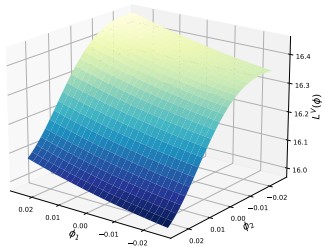

*Figure 12.* Value objective $L^V(\phi)$ is always smooth in the network parameter $\phi$.

for some constant $K_1 > 0$ where $\beta_1$ is the learning rate of policy network. We can also find another constant $K_2 > 0$ such that the difference in the value network output can be estimated by

$$|V(s; \phi_{k+1}) - V(s; \phi_k)| \simeq K_2 K_V \beta_2.$$

where $K_V$ is the number of optimization steps for the value network and $\beta_2$ is the learning rate. Therefore, the step sizes should satisfy

$$K_V \geq \frac{K_1 \beta_1^\alpha}{K_2 \beta_2} \tag{17}$$

so that the value network continues to provide accurate estimates of the true return and we complete the proof.

## D. Monte-Carlo Value Estimation

Most on-policy algorithms, including VPG, TRPO, and PPO, estimate the return value using either Monte Carlo methods or their generalizations (e.g., GAE (Schulman et al., 2015b)). However, the return estimated by Monte Carlo methods can only reflect how good (or bad) the sequence of actions $\mathbf{a} = (a_0, a_1, \ldots, a_T)$ is as a whole, given the initial state $s_0 \sim \rho$. Consequently, this return may deviate significantly from the true value $V^\pi(s_0)$ due to: (a) randomness in the stochastic policy $\pi_\theta$, and (b) exponential divergence of perturbed trajectories in the (potentially) chaotic dynamics.

**Mollified value landscapes.** It has been demonstrated that policy gradient methods work by smoothing the value landscape through a Gaussian kernel, thereby providing a valid updating direction even in the presence of fractal structures within the value landscape (Wang et al., 2024). While the averaged return can point to a correct direction, it also implies that a single trajectory is not sufficiently informative. For instance, consider two sampled trajectories with action sequences $\mathbf{a} = (a_0, a_1, \ldots, a_T)$ and $\mathbf{a}' = (a'_0, a'_1, \ldots, a'_T)$, where $a_0 \simeq a'_0$. The resulting returns, $G$ and $G'$, could be entirely different despite the initial actions being nearly identical. This discrepancy arises because subsequent actions may differ. In such cases, $\hat{A}(s_0, a_0)$ and $\hat{A}(s_0, a'_0)$ could have opposite signs, even when $a_0 \simeq a'_0$. Therefore, it underscores the importance of relying on the averaged return rather than focusing too much on individual trajectories.

**Understanding the Role of Baselines** Theoretically, both REINFORCE (3) and VPG (4) provide the same gradient estimator since

$$\hat{\mathbb{E}}_{(s_t, a_t) \sim \pi_\theta} \left[ \nabla \log \pi_\theta(a_t|s_t) Q_t \right]$$
$$= \hat{\mathbb{E}}_{(s_t, a_t) \sim \pi_\theta} \left[ \nabla \log \pi_\theta(a_t|s_t)(A_t + V^\pi(s_t)) \right]$$
$$= \hat{\mathbb{E}}_{(s_t, a_t) \sim \pi_\theta} \left[ \nabla \log \pi_\theta(a_t|s_t) A_t \right]$$

under the assumption that

$$\hat{\mathbb{E}}_{a_t \sim \pi_\theta} \left[ \nabla \log \pi_\theta(a_t|s_t) \right] = 0 \tag{18}$$

for any $s_t$ and any probability density function $\pi_\theta$. However, Equation 18 is not guaranteed in practice, where the number of samples may be insufficient to support it. Here, we demonstrate that the number of samples required to ensure the empirical mean satisfies $|\hat{\mu}| = |\nabla \log \pi_\theta(a_t|s_t)| \leq \epsilon$ becomes prohibitively large when $\epsilon$ is very small: Without loss of generality, let $\pi_\theta$ be a one-dimensional Gaussian distribution, i.e., $a \sim \mathcal{N}(\mu(s), \sigma^2)$. Suppose that $a_1, ..., a_K$ are i.i.d. samples obtained from $\pi_\theta$. The empirical mean

$$\hat{\mu}_K = \frac{1}{K} \log \pi_\theta(a_i|s) = -\frac{1}{K} \sum_{i=1}^{K} \frac{(a_i - \mu(s))^2}{\sigma^2}.$$

This implies that $-\hat{\mu}_K \sim \chi^2(K)$. Therefore, for each state $s_t$, approximately $\mathcal{O}(K)$ samples are needed. This requirement results in a total of $\mathcal{O}(K^T)$ samples per iteration.

This approach is exhaustive, as it involves exploring sufficiently many actions for each state encountered at every timestep. Consequently, it converges to tabular search approaches, offering no reduction in computational complexity. However, such exhaustive exploration is clearly not the intended purpose of policy gradient methods. In fact, this process resembles the *vine TRPO* algorithm, which partially expands the tree due to computational constraints (Schulman et al., 2015a).

It is worth noting that we have disregarded the accumulation of errors arising from the hierarchical structure. If these errors were taken into account, the required number of samples would be even larger, but we do not elaborate further on this point. Therefore, the assumption $\hat{\mathbb{E}}_{a_t \sim \pi_\theta} \left[ \nabla \log \pi_\theta(a_t|s_t) \right] = 0$ for any $s_t$ is too strong to be realistically satisfied in practice. This implies that using a baseline can significantly influence the performance of the gradient estimator. Specifically, when there is no baseline value function or when the baseline function is poor, the policy gradient estimator tends to converge toward suboptimal solutions. This behavior explains the poor performance of the original VPG algorithm in Gymnasium.

