# OpenReview forum: "Improving Value Estimation Critically Enhances Vanilla Policy Gradient"
_ICML.cc/2025/Conference — ICML 2025 poster_

### Official Review · Reviewer_rHz8 · 2025-03-13

**Overall Recommendation:** 4

**Summary:**

This paper demonstrates, through both theoretical analysis and experiments, that vanilla policy gradient (VPG) can achieve performance comparable to PPO by simply increasing the number of value updates per iteration. Contrary to the common belief and existing claims, the authors argue that accurate value estimation is more critical than enforcing trust regions. They provide theoretical justification and experimental validation to support this claim.

**Claims And Evidence:**

The claims regarding the importance of accurate value estimation are supported by empirical results. However, the assertion that trust regions (“trust regions are neither sufficient nor necessary for guaranteed performance improvement”) requires further theoretical backing.

**Essential References Not Discussed:**

None.

**Ethical Review Concerns:**

None.

**Experimental Designs Or Analyses:**

The experimental design is appropriate and the results are convincing. Broader evaluations across more algorithms (TRPO) and environments (e.g.,dm_control https://github.com/google-deepmind/dm_control) would strengthen the conclusions.

**Methods And Evaluation Criteria:**

The methods and evaluation criteria are appropriate. Including experiments with PPO and TRPO would provide a more comprehensive evaluation.

**Other Comments Or Suggestions:**

- Highlight the baseline analysis from the appendix more clearly in the main text.
- Clarify any safeguards used to prevent potential overfitting in iterative value estimation.

**Other Strengths And Weaknesses:**

**Strengths**

- The paper presents a compelling insight that accurate value estimation, rather than trust region enforcement, is key to policy improvement in policy gradient methods.
- The theoretical and empirical analysis is clear and well-structured, particularly the explanation of how TRPO and PPO implicitly enhance value estimation by performing more value updates per iteration.
- Experimental results seem strong, with thorough evaluations across environments.
- The findings have practical significance, suggesting that simpler algorithms like VPG, when combined with improved value estimation, can perform comparably to more complex methods such as PPO.

**Weaknesses**
- The experiments focus mainly on VPG with enhanced value estimation, without evaluating how the proposed approach performs in conjunction with PPO and TRPO. A direct comparison would strengthen the claims.
- Some statements, such as "trust regions are neither sufficient nor necessary for guaranteed performance improvement," are quite strong and would benefit from additional theoretical justification beyond empirical observations.
- The potential downsides of iterative value estimation, such as overfitting or diminishing returns, are not sufficiently addressed. Clarification on this would improve the paper.

**Questions For Authors:**

Q1. How does the proposed method perform when combined with PPO and TRPO rather than VPG? A direct comparison would clarify the general applicability of your findings.

Q2.  Are there situations where additional iterations of value estimation lead to degraded performance? How do you prevent overfitting or instability?

Q3. Will you release the complete implementation code and hyperparameter configurations to ensure reproducibility?

**Relation To Broader Scientific Literature:**

The paper makes a novel contribution by reframing the role of value estimation in policy gradient methods. It builds well on existing work policy gradient/optimization literature.

**Theoretical Claims:**

The theoretical analysis of baselines and sample complexity is sound. Additional theoretical discussion on the necessity of trust regions would be valuable.

---

> ### Author Rebuttal · Authors · 2025-03-27
>
> We would like to thank the reviewer for their positive and supportive review!
>
> > However, the assertion that trust regions (“trust regions are neither sufficient nor necessary for guaranteed performance improvement”) requires further theoretical backing.
>
> We will revise the statements in the paper to more accurately reflect both the current theoretical analyses and the empirical results.
>
> > Including experiments with PPO and TRPO would provide a more comprehensive evaluation.
>
> Please see the answer to Q2 below.
>
> > Broader evaluations across more algorithms (TRPO) and environments would strengthen the conclusions.
>
> That's a good point. We had considered running additional experiments in the DM Control Suite but finally decided against it after realizing that almost all on-policy algorithms perform poorly in that setting. For example, as shown in Appendix C.2 of https://arxiv.org/abs/2501.16142, PPO struggles in these environments. In contrast, only off-policy algorithms like SAC and TD-MPC consistently achieve strong performance.
>
> Regarding the weaknesses:
>
> 1. We agree that further investigation into enhancing value estimation in PPO is a promising direction for future work. Please refer to the answer to Q1 below for a more detailed response.
>
> 2. We will revise the statements in the paper to more accurately reflect both the current theoretical analyses and the empirical results.
>
> 3. Please see the answer to Q2 below.
>
> And for the questions:
>
> Q1: While this paper focuses on the pivotal role of value estimation in policy gradient methods, as highlighted in the title, this question is one of the directions we plan to explore in future work. It will be interesting to investigate how much PPO can benefit from enhanced value estimation.
>
> In the default PPO setting considered in this paper, the value network is updated with 320 gradient steps per iteration, which is generally sufficient for good value estimation. However, since PPO optimizes both the policy and value networks within a single loop, the same number of steps is applied to the policy network. This results in the policy network being over-updated, which contributes to the poor performance observed in the Ant and Humanoid tasks.
>
> Therefore, one possible direction is to reduce the number of optimization steps in PPO, as we have already shown that 50 value steps are sufficient for good value estimates. Additionally, the empirical results in this work indicate that even a single policy step per iteration can lead to good performance, suggesting that we can also reduce the number of policy steps in PPO. In Figures 9(c) and (d) in the Appendix, we demonstrate that a PPO implementation with 8 gradient steps per iteration significantly outperforms the default PPO implementation on the Ant and Humanoid tasks. These results provide concrete evidence that this approach could be a promising direction.
>
> We believe that by reducing the number of optimization steps, PPO can become more wall-time efficient and more robust to hyperparameters.
>
> Q2: That is an insightful question! In Figure 4, we run the VPG algorithm with varying numbers of value steps, which demonstrates that having the maximum number of value steps per iteration does not always result in the best performance. Although no significant performance decline is observed as the number of value steps increases, we found that 50 value steps per iteration yields the optimal final return across five different environments. Additionally, we did not observe any instability even with an increased number of value steps.
>
> The key insight we aim to highlight through this work is that enhanced value estimation can significantly improve the simplest VPG algorithm. The empirical results presented in Figure 4 suggest that even for those with limited experience in hyperparameter tuning, using a sufficiently large number of value steps can yield decent results. However, we agree that incorporating additional techniques to prevent overfitting could further enhance the robustness of VPG. In particular, since optimizing the value network is essentially a supervised learning problem, methods like early stopping and regularization, commonly used in deep learning, could be beneficial.
>
> Q3: Of course! The code for the experiments will be released once the paper is accepted.
>
> Thank you again! We will revise the paper accordingly based on your suggestions.

---

> > ### Comment · Reviewer_rHz8 · 2025-04-03
> >
> > Thank you to the authors for their response.
> > As pointed out by reviewer pFVP, the CleanRL library provides benchmark results for on-policy algorithms such as PPO and its variants, including continuous tasks like those in dm_control. The experiments in this paper can be extended to these settings, where on-policy algorithms typically perform well. Doing so would strengthen the paper’s claims, as the general assumption is that the findings of Value estimation in VPG should generalize across all standard continuous control tasks.

---

> > > ### Author Response · Authors · 2025-04-03
> > >
> > > Thank you for the suggestion. We agree that additional experiments in DM Control would further strengthen our conclusions, and we will include them in the paper.

---

### Official Review · Reviewer_pFVP · 2025-03-13

**Overall Recommendation:** 2

**Summary:**

The authors provide a novel perspective how the approximation error of the value function determines the performance of the algorithm more so than the proximal constraint in TRPO and PPO. they argue this both empirically and theoretically. They further argue that the accuracy of the value function adds to robustness against perturbations of hyper-parameters.

**Claims And Evidence:**

I have the following issues with the claims:

1. Right hand column line on page 1:" The gap between theory and practice arises because the theory guarantees improvement of the original policy objective only when the surrogate objective improves by a margin greater than the distance between the old and new policies." This is very strong. There are numerous gaps between theory and practice. I believe you must cite some other work or demonstrate this.

2. You claim that the objective is not differentiable and reference Wang et al 2023. I would think that the landscape becomes a lot smoother under the over-parametrized neural network's regime. This is because NNs implicitly regularize. Also, see References [1,2] below, the loss landscape is "smoother" upon being projected to low-dimensional space.

3. **Theorem 5.2** I would recommend the authors explain what they mean by "Assume that the dynamics, reward function" being Lipschitz continuous in theorem 5.2. This is especially important since this is a stochastic setting. It would benefit the reader by helping them understand the setting. Without that the claims of theorem 5.2 seem slightly difficult to grasp. Could you please explain why you end up with $\beta_2$ in the denominator by means of a brief proof sketch? I believe that would help understand the claim "This result suggests that the value network should be optimized more aggressively than the policy network".

4. I am not sure if the results in Figure 4 are very convincing to the larger point you make. I believe PPO is more sample efficient as compared to VPG-repeat-k methods. I would at least add smoothing to the curves to make the figures more interpretable, as it is standard practice.

----------------------------
References
----------------------------
1. A framework for training larger networks for deep Reinforcement learning, Ota, Kei and Jha, Devesh K and Kanezaki, Asako, 2024

2. Visualizing the Loss Landscape of Actor Critic Methods with Applications in Inventory Optimization. Recep Yusuf Bekci and Mehmet Gumus, 2020

**Essential References Not Discussed:**

I believe the following references should be discussed:

1. Return-based Scaling: Yet Another Normalisation Trick for Deep RL, Tom Schaul, Georg Ostrovski, Iurii Kemaev, Diana Borsa, 2021: this paper relates to reward scaling and should ideally be discussed in Section 6.

2. CleanRL: High-quality Single-file Implementations of Deep Reinforcement Learning Algorithms, Shengyi Huang, Rousslan Fernand Julien Dossa, Chang Ye, Jeff Braga, 2021, https://github.com/vwxyzjn/cleanrl this codebase deals with issues of doing a single loop for both value and policy networks and I ould expect the authors to comment on this.

**Experimental Designs Or Analyses:**

I see the following issues with experiment designs:

1. Do you run the experiment on a single seed for Figure 2? This might not be conclusive although it it indicative. A broader set of experiments for various environments might be necessary in this case.

2. "VPG should have performed well, as it always applies smaller learning rates compared to TRPO and fewer policy steps per iteration compared to PPO" -> What about large gradient magnitudes?

3. I am unsure as to how you compute $L^{SL}$. What is the dependence of r on $\theta$? Is it through the policy? Further, how do you sample for estimating this ratio and what is the true value $r*$?

4. "The TRPO implementation, however, requires separate optimization loops " -> I believe this issue was addressed for PPO and SAC by the cleanRL implementation and this should be discussed in addition to analyzing the implementation given its popularity and usage.

5. I am not sure if the results in Figure 4 are very convincing (see above).

**Methods And Evaluation Criteria:**

I believe the biggest drawback in the methods are the single seed experiments in Figure 2.

**Other Comments Or Suggestions:**

Minor issues:
1.  According to (Sutton et al., 1999) -> According to Sutton et al., 1999, line 85 column 2
2. Line 135 col 2: I find it hard to believe that the landscape for over-parametrised NNs is fractal. Could you provide some other reference than Wang et al 2023? while I believe that for a two parameter problem these are highly irregular but it might not be the case for NNs which implicitly smoothen the landscape.
3. In the "Ratio trust regions in PPO" section and "Trust region methods" section you use $r_t$ for probability ratio I would recommend using some other symbol because $R$ is used for reward and this can be a bit confusing.
4. Section 5.1: Please provide the reference for GAE here.
5. Section 5.2: "Similarly, having a properly optimized value network is also" I would not start a new subsection with "Similarly".

**Other Strengths And Weaknesses:**

See above.

**Questions For Authors:**

See above.

**Relation To Broader Scientific Literature:**

I believe this is an important line of work empirically exploring the shortcomings of deep RL methods and rectifying them.

**Theoretical Claims:**

I did not verify the correctness of theorem 5.2.

---

> ### Author Rebuttal · Authors · 2025-03-27
>
> Thank you for the important review. Hope that our response adequately addresses your concerns.
>
> For claims:
>
> 1. This argument directly corresponds to the Theorem 1 from the TRPO paper, which states that
> $$J(\theta) \geq L^{TRPO}(\theta) - \frac{4 \beta \gamma}{(1 - \gamma)^2} D_{KL}^{max}(\pi_\theta \ \| \ \pi_{\theta_{old}})$$
> where $J(\theta)$ is the original policy objective, $L^{TRPO}(\theta_{old})$ is the surrogate objective and $D_{KL}^{max}$ is the KL distance between the old and new policies. Since it always has $J(\theta_{old}) = L^{TRPO}(\theta_{old})$. Therefore, the original policy objective is guaranteed to improve only if
> $$J(\theta) - J(\theta_{old}) \geq L^{TRPO}(\theta) - L(\theta_{old}) - \frac{4 \beta \gamma}{(1 - \gamma)^2} D_{KL}^{max}(\pi_\theta \ \| \ \pi_{\theta_{old}}) \geq 0$$
> which requires that the surrogate objective is improved by a margin greater than $\frac{4 \beta \gamma}{(1 - \gamma)^2} D_{KL}^{max}(\pi_\theta \ \| \ \pi_{\theta_{old}})$. While the theory proves the above inequality when two policies are close enough, there is no way to guarantee that the second part of the inequality above always holds in practice.
>
> We will add the citation there and rephrase the statement as follows for clarification:
>
> > According to the theory of trust regions (cite TRPO paper here), there is a gap between theory and practice because the theory guarantees improvement of the original policy objective only when the surrogate objective improves by a margin greater than the distance between the old and new policies.
>
> 2. The landscape is fractal due to the chaotic dynamics of the MDP, which are independent of the policy parameterization. The loss landscapes in [1, 2] correspond to the actor loss over a given batch of data, rather than the return landscape shown in Figure 1. For further reference on fractal landscapes, please refer to [1].
>
>
> 3. That is a good point. This paper focuses on continuous control problems where the MDP is deterministic. The only source of randomness comes from the Gaussian policy we use, which is why we can make the Lipschitz assumption on the dynamics and reward. The proof of Theorem 5.2 can be found in Appendix C. The intuition behind having $\beta_2$ in the denominator is that smaller learning rates require more value steps to compensate.
>
> 4. We agree that PPO is more sample efficient than VPG, as confirmed in the second paragraph of the Conclusion. However, this comes at the cost of reduced stability, as demonstrated in the Ant and Humanoid results in Figure 4, where VPG-repeat-k significantly outperforms PPO when k is greater than 50. When referring to performance, we mean the final performance of each algorithm, and we will clarify this in the draft. We believe that the current empirical results are sufficiently convincing to show that VPG with enhanced value estimation can achieve comparable performance to PPO, as initially claimed.
>
> We will also smooth the reward curves.
>
> For experiments:
>
> 1. We ran that for a single seed as it was only an illustrative example, and we have re-run the Hopper task with multiple seeds for further validation. Please see our response to Reviewer duhH for the detail.
>
> 2. We hypothesize that larger gradient magnitudes used in TRPO may positively contribute to its performance improvement over VPG. However, if this were the case, it would contradict the core idea in the TRPO paper, which favors small policy updates. Additionally, it has been shown that larger learning rates often lead to poor performance [2]. Furthermore, in Section 6, we examine how learning rates influence performance, demonstrating that excessively large gradient steps can negatively impact the final return.
>
> 3. $r_t = \frac{\pi_\theta}{\pi}$ is exactly the probability ratio in PPO, so it depends on $\theta$, which is estimated in the same way as in PPO. $r_t^*$ is the target probability ratio, as defined in lines 170-171.
>
> 4. Yes, in lines 266-268, we mentioned that PPO uses a single optimization loop for both networks, which leads to ratio clipping that prevents the policy from over-updating.
>
> For the suggested references, we will add the following discussion accordingly to the main text:
>
> "It has been demonstrated that proper reward scaling can lead to overall performance improvements in deep RL [3]."
>
> "Like VPG, PPO uses a single optimization loop for both networks, which can be found in commonly used implementations such as CleanRL [4]."
>
> We will also revise the paper to address the minor issues you suggested.
>
> [1] Rahn et al., Policy Optimization in a Noisy Neighborhood: On Return Landscapes in Continuous Control, NeurIPS '23.
>
> [2] Andrychowicz et al., What matters in on-policy reinforcement learning? a large-scale empirical study, ICLR '21.
>
> [3] Schual et al., Return-based Scaling: Yet Another Normalisation Trick for Deep RL, 2021.
>
> [4] Huang et al., CleanRL: High-quality Single-file Implementations of Deep Reinforcement Learning Algorithms, JMLR '22.

---

### Official Review · Reviewer_aCm7 · 2025-03-14

**Overall Recommendation:** 3

**Summary:**

This paper studies the performance gap between vanilla policy gradient and PPO/TRPO type of trust region enforcing algorithms. The authors conclude through empirical studies that the core to the performance gap lies in value estimation, as opposed to the common belief of trust region enforcing.

**Claims And Evidence:**

Yes

**Essential References Not Discussed:**

No.

**Experimental Designs Or Analyses:**

Yes

**Methods And Evaluation Criteria:**

Yes

**Other Comments Or Suggestions:**

None.

**Other Strengths And Weaknesses:**

### Strengths:
Overall this paper is a clear empirical report on the performance of VPG as compared with region enforcing optimization algorithms. Especially clear and convincing is the empirical studies on the correlation between value estimation error and performance of vanilla policy gradient (figure 4 and 5).
### Weaknesses:
PPO performance is known to be highly sensitive to hyperparameters and mini-batch sizes [1]. It is unclear from the experiments performed that the performance dominance is from an improper choice of hyperparameters of PPO or the fact that VPG is with multiple update steps. It would also be more informative to compare the value estimation error of PPO against VPG.
Some prior work [2] has also done extensive study on such continuous control benchmarks and addressed that policy intialization actually plays an important role in the performance on-policy algorithms. It would be more extensive if the authors could provide insight on this observation as well.

[1] Paine, Tom Le, et al. "Hyperparameter selection for offline reinforcement learning." arXiv preprint arXiv:2007.09055 (2020).
[2] Andrychowicz, Marcin, et al. "What matters in on-policy reinforcement learning? a large-scale empirical study." arXiv preprint arXiv:2006.05990 (2020).

**Questions For Authors:**

All the benchmarks are conducted in continuous control settings, where the reward at each timestep is crucial, making value function estimation a core aspect of the problem. However, in domains such as language modeling and other discrete action settings, the action space tends to be significantly larger, and reward signals are often much sparser. Given these differences, it is unclear whether the same observations hold in these environments. The reliance on value function estimation may not translate as effectively, and the conclusions drawn from continuous control tasks may be more domain-specific than universally applicable, as the authors suggest.

**Relation To Broader Scientific Literature:**

An interesting argument for commonly used RL algorithms.

**Theoretical Claims:**

Yes

---

> ### Author Rebuttal · Authors · 2025-03-27
>
> We thank the reviewer for their positive feedback and hope that our response can address your concerns.
>
> Regarding the weaknesses:
>
> We used the PPO implementation from Tianshou with learning rate decay and reward normalization disabled to better compare algorithmic performance without excessive code-level optimizations. We also acknowledge that mini-batch size is a crucial factor in PPO's performance and have conducted an ablation study on this in Figure 9(b) in the Appendix.
>
> The value estimation error of PPO and VPG is compared in Figure 5, which shows that the default PPO implementation provides good value estimates, as it performs 320 value steps per iteration, which is sufficient for accurate estimation. We agree that policy initialization is crucial for on-policy algorithms, so we fixed the random seed to ensure that both PPO and VPG start from the same initial policy and value network.
>
> For the question:
>
> The question raised here is insightful. Our algorithm focuses on continuous control settings, where PPO is widely believed to consistently outperform VPG. However, in other domains, such as language modeling (LLM), even the simplest REINFORCE algorithm has been reported to be able to outperform PPO ([1]).
>
> We agree that despite being the standard benchmark in deep RL, the success in MuJoCo tasks does not guarantee its effectiveness in other domains. In this paper, both our theoretical analysis and empirical results are focused on continuous control and robot learning, as emphasized in the Conclusion. We will revise the paper to ensure that our claims are clearer and more specific to this context.
>
> [1] A. Ahmadian et al., Back to Basics: Revisiting REINFORCE Style Optimization for Learning from Human Feedback in LLMs, arXiv:2402.14740.

---

### Official Review · Reviewer_duhH · 2025-03-14

**Overall Recommendation:** 2

**Summary:**

This paper investigates the role of value estimation accuracy in on-policy policy gradient reinforcement learning. The authors demonstrate that improving the accuracy of value estimation (by performing more value function updates per iteration) dramatically improves the data efficiency of vanilla policy gradient methods. While it is widely accepted that TRPO and PPO perform well because they prevent destructively large policy updates, this paper suggests that value estimation plays a more critical role in their success than previously recognized.

**Claims And Evidence:**

The paper makes several key claims, which I evaluate below.

**Claim 1: Trust regions are neither sufficient nor necessary for guaranteed policy improvement.**

* **Partially supported.** The paper convincingly shows that the trust region is typically violated in PPO implementations, yet PPO still learns effectively. This supports the argument that enforcing a trust region is not necessary for policy improvement. However, the claim that trust regions are not sufficient for policy improvement is less convincing. The authors test this by adding a supervised loss (Eq. 9) to the PPO objective that encourages the policy to stay at the edge of the trust region. My concern is that this auxiliary loss may dominate the update and potentially drown out the contribution of the policy gradient, effectively preventing the policy from following the policy gradient direction. An alternative approach to test this claim might be to reduce the learning rate while setting the number of epochs to a much higher value, which would push the policy close to the trust region boundary while reducing trust region violations.

**Claim 2: TRPO and PPO implicitly enhance value estimation in their implementations.**

* **Theoretically supported but lacks empirical validation.** Theorem 5.2 establishes that increasing the number of value function updates yields more accurate value estimates under certain conditions. However, the paper doesn't directly demonstrate that TRPO and PPO improve value estimation by performing multiple value updates per iteration. To strengthen this claim, the authors should experiment with reducing the number of value function updates in TRPO and PPO to observe if this produces worse value estimates.

## Claim 3: By increasing the number of value update steps per iteration, vanilla policy gradient can achieve performance comparable to or better than PPO.

* **Partially supported.** Figures 4 and 5 demonstrate that more accurate value estimation improves the data efficiency of VPG substantially. However, the comparison to PPO appears problematic as the PPO implementation seems suboptimally tuned in these experiments. Prior work has shown that well-tuned PPO performs consistently across all MuJoCo tasks, while the results here show declining performance in environments like Ant and Walker. Nevertheless, the substantial improvements to VPG through enhanced value estimation are clearly demonstrated.

**Claim 4: Performance of a policy gradient algorithm is primarily determined by value function estimation accuracy rather than specific policy training mechanisms.**

* **Not adequately supported.** The paper doesn't establish a clear framework for comparing the relative importance of value estimation versus trust region mechanisms. To support this comparative claim, the authors would need to demonstrate that a given improvement in value estimation yields better performance than an equivalent improvement in trust region enforcement. Additionally, Figure 5 shows that VPG-repeat-N achieves smaller value estimation differences as N increases, yet PPO's value estimation differences are similar to VPG-repeat-500, suggesting this metric may not strongly correlate with performance. I recommend rephrasing this claim to focus on the overlooked importance of value estimation rather than asserting its primacy.

**Essential References Not Discussed:**

None.

**Experimental Designs Or Analyses:**

1. However, the experimental design would be stronger if it also examined increasing the number of value updates in TRPO and PPO implementations, not just VPG. If the paper's hypothesis is correct, TRPO and PPO should also benefit from more accurate value estimates, and demonstrating this would strengthen the paper's conclusions.

2. In the experiments for Figure 2: Looking at the max ratio here could be misleading. What does the distribution of ratios look like? What if only one ratio is large and the rest are very small? Is there a reason Figure 2 only shows one seed per method? Appendix A shows that experiments include gradient clipping -- this will also implicitly prevent large updates and should be removed for the purpose of analysis.

3. Do VPG, TRPO, and PPO normalize observations and rewards?

**Methods And Evaluation Criteria:**

1. The benchmarks and baseline algorithms are appropriate for the RL experiments.

2. In the experiments for Figure 2: Looking at the max ratio here could be misleading. What does the distribution of ratios look like? What if only one ratio is large and the rest are very small? Is there a reason Figure 2 only shows one seed per method? Appendix A shows that experiments include gradient clipping -- this will also implicitly prevent large updates and should be removed for the purpose of this analysis.

**Other Comments Or Suggestions:**

> Vanilla Policy Gradient (VPG, (Sutton et al., 1999)), typically performs significantly worse than the modern algorithms. The common explanation for why TRPO and PPO outperform VPG is their ability to prevent excessive large policy updates.

We can make these statements more precise: VPG is typically less *data efficient* than the modern algorithms. The common explanation for why TRPO and PPO outperform VPG is their ability to *perform large policy updates while preventing destructively large updates.*

VPG performs a single gradient update per batch of data collected and must use a small learning rate to prevent destructively large changes to the policy. TRPO and PPO have mechanisms that mitigate destructively large policy updates, so they can perform larger gradient updates (TRPO) and more gradient updates per batch collected (PPO), which yields more data efficient learning.

> the gradient ∇θJ(θ) may not exist for many continuous control problems in the first place.

Does this statement allude to how we may be trying compute the policy gradient at corner in the optimization surface?

> ...VPG can achieve performance comparable to PPO.

This statement could be more precise. Are we talking about final performance, data efficiency, etc?

> Figure 1

Are we plotting the loss surface for a randomly initialized policy? Does the optimization surface smooth out as the policy improves? Is \$theta$ a multi-layer network, a linear network, etc? TRPO works well empirically even if the loss landscape is fractal (e.g. TRPO can perform reasonably well on MuJoCo tasks like Hopper and Walker), so is this actually a problem?

> Figure 5

It is difficult to read this figure because of the white markers -- I suggest removing them.

> Notably, during PPO training, the maximum probability ratio maxt rt can significantly violate the trust region defined by the clipping coefficient ε (Engstrom et al., 2020). This occurs because the gradient ∇θrt vanishes outside the trust region (e.g., when rt > 1 + ε) due to the clipping operator. Consequently, if an excessively large step is taken on πθ(at|st) such that rt leaves the trust region, it may not be pulled back to the trust region anymore due to the vanishing gradient.

I think this issue may arise due to an implementation detail of PPO. In practice, PPO stops updating once we *violate* the KL threshold $\delta$ rather than *just before* we violate the threshold. To better enforce the KL constraint, PPO should roll-back the update that caused the policy to violate the threshold.

For future work, it would be interesting to run VPG/PPO/TRPO with exact value estimates to see just how much we could potentially improve over the base algorithms. If this is too computationally demanding for MuJoCo benchmark tasks, it would be valuable to study even in a simple grid world task, where value iteration could compute V(s) exactly. This would establish an upper bound on the potential gains from perfect value estimation across different algorithms.

**Other Strengths And Weaknesses:**

**Additional Strengths**
1. I think this is an interesting topic, and I have considered investigating it myself, so I'm happy to see this paper.

**Additional Weaknesses**
1. is VPG with 100-500 value updates per iteration a scalable algorithm? What is the computational cost of VPG vs. VPG-repeat-500 vs. PPO? To clarify, I think it's fine if the computation cost is large. I think it's useful to comment on the cost though. If it's not scalable, that might be an interesting direction for future work.

**Questions For Authors:**

**I lean to reject because of the concerns I raise in the following questions. If these questions are appropriately addressed, I will raise my score.**

1.  I was surprised to see PPO perform so poorly on Walker and Ant, since it is possible for PPO to perform well on all MuJoCo tasks (e.g., see the [PPO benchmark by Tianshou](https://tianshou.org/en/stable/01_tutorials/06_benchmark.html)). Is PPO tuned? Is VPG tuned? In appendix A, it looks like the same hyperparameters are used across all PPO experiments, but different VPG learning rates are used for each task, suggested PPO is not tuned while VPG is. If this is true, PPO should also be tuned if you want to fairy compare VPG with PPO. On that note, I want to mention again that I don't think VPG needs to outperform PPO to make this work valuable, though it would be valuable to show that improving value estimate also improves PPO performance, since PPO is so widely used.

2. Regarding Figure 2: Could you show the distribution of probability ratios across multiple seeds, rather than just the maximum ratio? While the results show that the maximum ratio significantly violates the trust region, this doesn't tell us whether such violations are widespread or isolated. If the vast majority of ratios still fall within the trust region bounds and only a small fraction violate it, then PPO may still be effectively maintaining approximate trust region constraints in practice, which contradicts claim 1. If a non-negligible fraction of ratios violate the trust region, then the stated claims are okay.

3. Regarding Figure 5: PPO and VPG-repeat-500 achieve similar value estimate differences, yet VPG performs much better. Why? This observation seems to contradict claims 3 and 4.

**Relation To Broader Scientific Literature:**

To my knowledge, the role of value estimation in the context of PPO has been understudied.

**Theoretical Claims:**

I briefly looked at the proof of Theorem 5.2. One question: Does Eq. 17 simply follow from a Taylor expansion?

---

> ### Author Rebuttal · Authors · 2025-03-27
>
> We appreciate the reviewer's thorough and insightful feedback.
>
> For claims:
>
> 1. In Figure 2(c)(d), we observe that the new loss with 50 epochs effectively enforces the trust region compared to runs with fewer epochs, despite its poorer performance. This coincides with the suggested alternative approach. Alternatively, we may adopt a weaker statement: trust regions do not necessarily need to be enforced for policy improvement.
>
> 2. In Figure 5 and 11, we observe a significant value estimation error in the original VPG implementation with only one value step per iteration. Therefore, reducing the number of value steps in TRPO and PPO would likely lead to similar results, as the value estimation module is identical across all on-policy algorithms.
>
> 3. See later responses.
>
> 4. This point is very interesting, and we have considered. Actually, we found that it is hard to say if an improvement in trust region enforcement is equivalent to a given improvement in value estimation. One possible metric is the number of optimization steps made to the policy and value networks in PPO. However, it is not appropriate, as Theorem 5.2 indicates that the policy and value networks evolve at different rates, meaning that applying the same number of gradient steps does not necessarily lead to equal improvements. Additionally, ratio clipping in the policy further complicates the comparison. While Theorem 5.2 provides a lower bound, determining the exact ratio of gradient steps that results in equivalent improvements in value estimation and trust regions is impossible. We will rephrase the draft accordingly based on your suggestions.
>
> For experiments:
>
> 1. In the default PPO setting used in our paper, the value network takes 320 gradient steps per iteration, which is already sufficient to provide strong value estimation, as shown in Figure 5.
>
> 2. We adopted the maximum ratio metric from Engstrom et al.. In Figure 2, only a single random seed was used, as it serves primarily as an illustrative example. However, we have re-run the Hopper task with five seeds for further validation.
>
> Regarding the empirical results, we observe that the mean fraction of ratios violating the trust region bounds increases from 0.19 to 0.32 through training, with a standard deviation of 0.01 across five random seeds using the original PPO implementation from Tianshou. A similar trend is also reported in Figure 3 of [1]. These observations support our claims, as a non-negligible fraction of ratios consistently violate the trust region.
>
> Gradient clipping is applied only to the experiments in Sections 5 and 6 and is no longer used in the illustrative examples in Section 4.
>
> 3. Only the observation is normalized.
>
> For weaknesses:
>
> In the default PPO setting, it takes 320 optimization steps per iteration, compared to 501 steps in VPG-repeat-500. However, Figure 4 shows that VPG-repeat-50 already outperforms PPO with just 51 steps. This suggests that the approach is scalable.
>
> For comments:
>
> 1. When referring to performance, we talk about the final return and will clarify this.
>
> 2. Figure 1 shows that the return landscape of TRPO is fractal and violates the assumptions made in the theory. However, despite this, TRPO still performs well in practice. This suggests that its empirical success may not be entirely explained by the theory of trust regions, which then motivates the findings presented in our paper.
>
> **For questions**:
>
> 1. For the experiments in Section 6, **we used the PPO implementation from Tianshou with learning rate decay disabled (disabled in VPG, too)**, which could explain the performance decline observed in Walker and Ant. We will update the hyperparameters in Appendix A to reflect these. We believe that anyone using this setting can reproduce the same results and will release the code afterwards.
>
> As analyzed in Section 4, ratio clipping in PPO is not sufficient to effectively enforce trust regions or completely prevent large policy updates. This limitation helps explain why the default PPO struggles to solve the Humanoid task. Additionally, random seeds may also contribute to the observed performance differences. Similar declines in performance on Ant are reported in Figure 17 of the ICLR '21 paper by Liu et al..
>
> Regarding the learning rate of VPG, we adopted 7e-4 from Stable-Baselines3. The 3e-4 was specifically chosen for Hopper is to allow for a better comparison with PPO on that task. **As shown in Figure 7(a), using 7e-4 in Hopper actually results in better performance.**
>
> 2. See previous responses.
>
> 3. Our results show that VPG can perform similarly to PPO with enhanced value estimation, which is the key contribution of this work. **However, with PPO, other factors also play a role. As previously mentioned, PPO's performance may decline due to its aggressively updated policy**, which explains why VPG-repeat-500 outperforms PPO despite both having similar value estimations.
>
> [1] Wang et al., Truly Proximal Policy Optimization, UAI, 2020.

---

### Decision · Program_Chairs · 2025-05-01

**Decision:**

Accept (poster)

**Comment:**

This paper challenges the common belief that trust region is the key factor for the good performance of modern policy optimization methods such as TRPO and PPO. The authors present a compelling argument, supported by both theoretical analysis and empirical evidence, that enhanced value estimation accuracy plays a more critical role. The show that vanilla policy gradient (VPG) can achieve comparable performance to PPO by simply increasing the number of value update steps.

From my read, I found the paper has several strengths. It provides a though-provoking perspective on the importance of value estimation in policy gradient methods. And the empirical studies are conducted in a reasonably convincing way to me.

However, reviewers also found that the paper has some weaknesses. There are valid concerns about the experimental validation (the tuning of PPO and the potential impact of hyperparameters) and generalizability of the findings (in continuous setting value estimation is an issue, which means the findings might not be generalized to other settings such as discrete and LLM related settings). Some reviewers also found that some claims are too strong and the experimental results only support partial or weaker versions of those claims.

Overall, the theoretical and empirical evidence presented is sufficient to warrant a weak accept. The authors should revise the paper to address the remaining concerns and clarify the scope of their claims.